# Qualitative Analysis Related to Palynological Characterization and Biological Evaluation of Propolis from Prespa National Park (Greece)

**DOI:** 10.3390/molecules27207018

**Published:** 2022-10-18

**Authors:** Elisavet Pyrgioti, Konstantia Graikou, Nektarios Aligiannis, Sofia Karabournioti, Ioanna Chinou

**Affiliations:** 1Lab of Pharmacognosy and Chemistry of Natural Products, Faculty of Pharmacy, National and Kapodistrian University of Athens, 15771 Zografou, Greece; 2Attiki Bee Culturing Co., 14568 Athens, Greece

**Keywords:** GC-MS, Prespa National Park, propolis, antioxidant properties, antibacterial, antifungal activity

## Abstract

Propolis samples from a geographical part of northwest Greece (Prespa National Park, PNP), which is characterized as a plant endemism center and biodiversity hotspot, were characterized through pollen analysis, chemically analyzed, and biologically evaluated. The majority of the studied propolis showed typical chemical constituents (phenolic acids, flavonoids, and chalcones) of European type, while a sample of Mediterranean-type propolis (rich in diterpenes) was also identified. The palynological characterization was implemented to determine the botanical origin and to explain the chemical composition. The total phenolic content and the DPPH assay showed that the European-type propolis samples possessed strong antioxidant activity (86–91% inhibition at 200 μg/mL). Moreover, promising antibacterial activity of the extracts (MIC values 0.56–1.95 mg/mL) and moderate antifungal activity (MIC values 1.13–2.40 mg/mL) were noticed, while the sample with the highest activity had a significant content in terpenes (Mediterranean type). Propolis samples from the PNP area represent a rich source of antibacterial and antioxidant compounds and confirm the fact that propolis is a significant natural product with potential use for improving human health and stimulating the body’s defense. Finally, it is noteworthy that a significant chemical diversity was demonstrated, even in samples from a limited geographical area as this of PNP.

## 1. Introduction

Propolis is a balsamic and resinous substance, produced by bees (*Apis mellifera* L.) using different plant exudates, which has been used in the hive as a building and sealing material, as well as to protect from invading predators, while also offering chemical defense against microorganisms, creating an aseptic internal environment [1]. The word “propolis” is of Greek origin and exactly reveals its role when translated: “before the city”, defining the way it works in the defense of the community and the beehive. Propolis is characterized by a variable chemical composition, as it is a complex mixture, depending both on the plant source of each geographic region as well as on the collected period [2,3]. It has been found that propolis can consist of more than 180 different types of chemical structures, while more than 300 different metabolites have been identified [4,5,6,7]. The valuable biological properties of propolis were recognized by humans since antiquity as its use is recorded in traditional medicine from ancient Egyptians, Romans, and Greeks [8]. Nowadays, propolis continues to be extremely commercially important in traditional medicine, showing many biological effects such as antioxidant, antibacterial, antimycotic, antiviral, antiparasitic, cytotoxic, immunomodulatory, anti-inflammatory, and hepatoprotective properties [7,9]. During the SARS-CoV-2 pandemic, propolis had intensified data, demonstrating its potential beneficial role through important antiviral properties, both in vitro and in silico, as well as its administration in clinical studies with promising preliminary results [10,11,12,13,14,15].

The Prespa Lakes are the highest tectonic lakes in the Balkans, characterized by rare local flora including protected species. There are two lakes separated by a narrow isthmus: Megali (Big) Prespa, which is located on the tripoint of Greece, North Macedonia, and Albania, and Mikri (Small) Prespa, which belongs mainly to Greece (approximately 43.5 sq. km.), while a tiny part (less than 4 sq. km.) to the west belongs to Albania. Both lakes (Big and Small Prespas) are located at altitudes of 852 and 857 m, respectively. The PNP of Greece covers an area of 257 km^2^ with its main characteristic being habitat diversity. The altitude range, from wetland habitat (altitude 850–853 m) to alpine meadows (altitude over 2000 m), has given rise to a species assemblage that is unique by international standards. Archeological records indicate that Prespa basin has been inhabited for more than six thousand years, while currently, on the Greek side of the region, the lakes are encircled by extensive reedbeds and farmland for the cultivation of beans. In 1975, the Prespa area was designated a “Landscape of Outstanding Natural Beauty”, and, in 2009, 11 smaller areas of the PNP were designated “Protected Natural Formation or Landscape Elements” [16,17]. Regarding the climate of this region, it can be typically characterized as Mediterranean or sub-Mediterranean with strong continental effects, due to its placement in a transitional zone between the Mediterranean climate in the south and continental climate in the north, and also due to the cold and snowy winters of the region and the summer rain seasons. The lands across the basin are characterized by geological diversity, revealing gneiss and granite on the slopes of Mt. Varnous and limestone on the mountains to the west and south side of the lakes [16,17].

PNP has an unusually high number of plant species per unit area and a high proportion of locally endemic species. The local flora consists of species unique for the Greek flora, species under international protection, and rare local species threatened with extinction, while a percentage of more than 15% of them represents endemic species. According to the mapping of habitat types of Directive 92/43/EEC in the Prespa area, 30 habitat types are found, while it is remarkable that in the area, a total of over 70 different types of vegetation have been recorded. Furthermore, until November 2011, more than 1800 species and subspecies of vascular plants have been recorded, of which 194 are considered particularly important [17,18,19].

In the Greek part of the PNP, there are only 5 apicultural centers with stable beehives, contrary to the bee-keeping common Greek practice of nomadic apiculture. In the present study, these five different samples of propolis from the PNP (NW Greece) have been determined through pollinic spectra quantitative microscopical analyses, studied chemically through GC-MS, and evaluated for their antibacterial, antifungal, and antioxidant capacity. To the best of our knowledge, there is no previous study on any kind of bee-keeping product from this geographic area.

## 2. Results

### 2.1. Propolis Composition

#### 2.1.1. GC-MS Analysis

The chemical composition of the studied propolis samples (70% ethanolic extracts) was investigated by GC-MS, after silylation (Table 1).

The analysis according to the chemical categories (Table 2) demonstrated in four of the samples (PP1, PP2, PP4, and PP5) the existence of phenolic acids (mainly ferulic and caffeic acid and their derivatives), flavonoids, and chalcones (such as pinocembrin, pinobanksin, and pinostrobin chalcone), classifying them as European type, while in one sample (PP3), the abundant compounds were the diterpenes (pimaric acid, communic acid, and ferruginol), classifying it as Mediterranean-type propolis [1,20].

#### 2.1.2. Isolation of Compounds

Among the studied samples, seven metabolites: pinocembrin, sakuranetin, pinostrobin, 7-oxo-dehydroabietinol, dehydroabietic acid, *Ε*-communic acid, and *Z*-communic acid, were isolated and their structure was determined by spectral data by comparison with the literature.

### 2.2. Pollen Grain Analysis

Pollen grain analysis (Table 3) showed five different families as secondary pollen (16–45%) and four families for the important minor pollen (3–15%).

Regarding samples PP1 and PP2, the predominant pollen grains correspond to *Centaurea* sp. (Asteraceae) and species of the genus *Pyrus* (Rosaceae). The presence of common pollen grains in these samples was expected, as both came from the same stable beehives, with the only quantitative difference being the fact that they were harvested in two different collection years, and the existence of different plant species in each year of collection was detected. In sample PP1 (collection season: summer 2019), *Centaurea* sp. (27%) was the abundant pollen grain followed by Liliaceae (25%) and Rosaceae (24%), whilst in sample PP2 (collection season: summer 2020), Rosaceaea and Fabaceae were the abundant families (25% each), followed by Asteraceae (16%). The sample deriving from the area of Vrondero (PP3) demonstrated similarities to sample PP1 in the presence of Liliaceae (26%) and Fabaceae (24%), but also a significant difference in the occurrence of pollen grains from Boraginaceae (22%). Finally, samples PP4 and PP5 also contained plants from the Fabaceae family at a percentage of 25 and 27%, respectively. In addition, the PP4 sample showed pollen grains of the Liliaceae family (28%), whilst the PP5 sample of the Asteraceae family showed 28%.

Analysis of nectarless plants (Table 3) revealed that all samples contained pollen grains from *Quercus* sp. (Fagaceae), and all of them, apart from sample PP3, contained *Pinus* sp. (Pinaceae). Sample PP3 contained a nectarless pollen grain from the Cupressaceae family, probably from *Juniperus* due to the internationally unique “cedar” forest, located near to the beehives area (Vrondero).

### 2.3. Antimicrobial Activity

All the ethanolic extracts of the propolis studied samples (PP1–PP5) were evaluated for their antimicrobial activity by the diffusion and dilution method against eight Gram-negative and -positive bacterial strains and three human-pathogenic fungi. The results of these tests (Table 4) showed significant and promising antibacterial activity (MIC range values from 0.56 to 1.95 mg/mL), with the PP3 sample showing the best among all, as well as moderate antifungal activity (MIC range values from 1.13 to 2.40 mg/mL).

### 2.4. Total Phenolic Content and Antioxidant Activity

The total phenolic content (TPC) of propolis extracts was determined by the Folin-Ciocalteu method [21]. The results of the assay (Table 5) showed a rich phenolic content for European-type propolis samples (153–203 mg GAE/g of dry extract), while sample PP3 showed the lowest content (39 mg GAE/g of dry extract).

The DPPH radical scavenging activity of the extracts showed a high inhibition for European-type propolis samples (86.74–91.31% inhibition at 200 μg/mL) (Table 5) and a low inhibition for the Mediterranean type of propolis, which follows accordingly the total phenolic content.

## 3. Discussion

The chemical profile of the propolis samples, as captured by the GC/MS method, is related to the plant origin of the palynological characterization. Flavonoids and aromatic acid esters, identified in the majority of the studied samples, are well known in general of *Centaurea* sp. (Asteraceae) and *Trifolium* sp. (Fabaceae) [20,22,23], which are the main resources of pollen grains in the studied species. Furthermore, the flavonoids (pinocembrin, pinobanksin and its 3-*O*-acetate, chrysin, and galangin) and aromatic acid esters (phenylethyl caffeates) are characterized as typical constituents for the bud exudates of *Populus nigra*, representing the European type of propolis [20].

On the other hand, the classification of sample PP3 as Mediterranean type was due to the presence of diterpenes. At first sight, this finding sounded strange due to the geography of PNP per se, but could be further explained, as the main plant source of diterpenes is mostly the coniferous trees of the Cupressaceae and Pinaceae families [1]. The PNP area is a famous source of five taxa of *Juniper* (Cupressaceae), among which the most common is *Juniperus communis* subsp. *communis*, which is pervasive in the park, specifically on the rough pasture lands on Mount Triklario (Sfika) above 1400 m and Mount Varnous at 1300–1800 m, and it is remarkable that woodlands of the tree junipers are rare in Southeastern Europe [18]. On the other side, the riparian forest, along the river above Agios Germanos, consists of willows (*Salix alba* and *Salix rubens* -Salicaceae) [18], which could also explain the presence of flavonoids in these samples. 

In addition, on Mount Devas, near the summit (1200–1300 m), there is a presence of open woodland of *Quercus trojana* (Fagaceae) and *Juniperus* on the rocky limestone slopes. This area near Vrondero (2.5 km E of Vrondero, 1280–1300 m) could explain the presence of diterpenes in sample PP3 [18].

Isolation of the 7-oxo-dehydroabietinol (sample PP3) from propolis for the first time led to the further investigation of its plant source. This metabolite has been mentioned before in *Pinus* and *Juniperus* species [24,25]. After GC/MS analysis of different *Juniperus* species from PNP, it has been proven that *Juniperus oxycedrus* L. subsp. *deltoides* from the same region was the source of this compound. Furthermore, pinocembrin has been identified in several propolis samples, mainly coming from *Pinus* trees [1,26,27] as well as the sakuranetin in Brazilian propolis from *Prunus* species (*Prunus jamasakura* and *Prunus verecunda*) [28]. Pinostrobin has been isolated from *Pinus strobus* (L.) and other *Pinus* sp., as well as from *Populus* sp., and it is a common compound in propolis of poplar-type [5,29]. Dehydroabietic acid has been identified also in *Pinus* [30] and in propolis samples of Mediterranean-type [1] as well as *E*- and *Z*-communic acids characteristic metabolites of *Juniperus* [31], which are also common in Mediterranean-type propolis [3,32].

The pollen analysis showed that the abundant plants belong to nectariferous plants (Apiaceae, Asteraceae, Boraginaceae, Liliaceae, Fabaceae, and Rosaceae), although there are also nectarless sources available. Specifically, *Centaurea* sp. seems to be generally preferred by bees, whilst beekeepers value it as a source of foraging at the end of summer [33], and it is worth mentioning that the genus *Centaurea* is widespread in the Prespa area, while the recent recording of two new species is indicative of the richness of the area: *C. galicicae* and *C. tomorosii* [18]. 

Rosaceae is one of the main families in most of the studied samples, as *Prunus* and *Pyrus* are widespread in the area, confirming the fact that this family is present in the main types of unifloral honey produced in Greece (chestnut, fir, pine, orange, cotton, thyme, and honey) [34]. 

Additionally, plants of the genus *Trifolium* (Fabaceae) are widespread in the area, while the Boraginaceae family (referred in sample PP3) is justified by the fact that species of this family (*Cynoglossum* sp.) grow close to the apiculture area [18].

Respecting the nectarless plants, it was observed that *Quercus* sp. (Fagaceae) appears in all the samples, which is explained by their presence of oak trees in the area close to the sample collection. *Pinus* sp. (Pinaceae) appears in all the samples out of PP3, due to the existence of forests of pine trees in the area at an altitude of less than 2000 m [18], while *Pinus* sp. is also referred in a study of pollen-derived sediments, thus demonstrating their long-term presence in the region, as well as their prevalence [35].

It is noteworthy that an increased percentage of glycerol was observed in the majority of the studied samples, without being expected according to the known chemical composition of propolis. The reason of this increased percentage is a beekeeping technique concerning the use of oxalic acid strips (with glycerol) for the treatment against *Varroa* mites, which is a genus of mites that live by parasitizing bees. Given the widespread use of these films (either prepared by beekeepers or commercially available), these high percentages seem to be fully justified. Bibliographically, this technique is highly effective, with advantages concerning the safe handling by beekeepers, while at the same time, no side-effects seem to be recorded for bees and colonies. As for the bee products, it is recorded that both honey and wax are not affected by this technique [36]. However, in the case of propolis, it has to be highlighted that the use of this technique could affect the chemical composition of propolis and, therefore, its use should be avoided in cases where beekeepers wish to commercially exploit it.

The antimicrobial activity of the European-type propolis (PP1, PP2, PP4, and PP5) could be attributed mainly to the content of flavonoids such as galangin, pinocembrin, and pinobanksin, which are known to possess high antimicrobial (antibacterial as well as fungicidal) activity, while the Mediterranean-type sample (PP3) with the stronger effect had a significant content in diterpenes, which lead to high antimicrobial activity according to the literature [27].

The DPPH radical scavenging activity of the extracts showed a high inhibition for European type propolis samples and a low inhibition for the Mediterranean type of propolis, which follows accordingly the total phenolic content. These results confirm the data from previous studies [3], in which propolis rich in flavonoids and phenolic acids and their esters possess strong antioxidant activity. On the other hand, Mediterranean propolis samples [1] with significant amounts of diterpenes have shown a low quantity of phenolic acid esters and, therefore, a lower antioxidant activity.

## 4. Materials and Methods

### 4.1. Samples

Propolis originated from stable comb hives was collected from five distinct areas in the PNP region, located in Northwest Greece (Figure 1, Table 6). The five samples were collected from Agios Germanos (PP1 and PP2) (2019 and 2020), Vrondero (PP3), Latsista (PP4), and Agios Achillios (PP5) during the summer period. The term stable beekeeping describes the process in which the beehives are in specific distinct positions without moving between different seasons unlike nomadic Greek beekeeping common practice. Therefore, the chemical composition of the samples is not affected by different plant origin in the same seasonal harvests, but only by the different climatic conditions of the area.

### 4.2. Pollen Analysis

The pollen analysis in propolis was carried out according to the method described by Ricciardelli D’Albore [37]. Here, 200 mg of propolis was dissolved and homogenized in a solution of ethyl alcohol, chloroform, and acetone (1:1:1). The centrifuged solution and the obtained sediment were dissolved in 20 mL of 10% KOH. The mixture was then boiled for 2 min, centrifuged, and collected with 10 mL of ethyl alcohol. The sediment was centrifuged and separated for a second time. The entire amount of sediment was transferred onto a slide. A disposable plastic Pasteur pipette was used to transfer the sediment from the centrifuge tube to the slide. The sediment was dried with gentle heating, not above 40 °C, and was covered with Entellan (Merck). The identification and quantification of pollen grains were carried out as in honey [38]. To present the frequency as a percentage (%), at least 500 pollen grains were counted from two slides created separately from the same sample. Some pollen grains could not be identified to the species level. In these cases, pollen grains could be classified at the genus or family level. Pollen grains, depending on their frequency, were classified into the following categories:”Predominant pollen” (more than 45% of the pollen grains counted); “Secondary pollen” (16–45%); “Important minor pollen” (3–15%); “Minor pollen“ (less than 3%). Pollen whose frequency was less than 1% was characterized as “isolated”. Pollen grains from nectarless plants were simply recorded as present.

### 4.3. Extraction and Sample Derivatization

Propolis was extracted three times with 70% ethanol (1:10, *w*:*v*) by immersion at room temperature for 24 h each time, followed by filtration of the resulting suspension at room temperature using a paper filter and, in vacuum, evaporation of the solvent to dryness on a rotary evaporator. For GC/MS analysis, the ethanol extracts were silylated (5 mg of each residue with 40 µL of dry pyridine and 50 µL of BSTFA (bis[trimethylsilyl] trifluoracetamide) and heated at 80 °C for 20 min) [1].

### 4.4. GC-MS Analysis

The component analysis was performed by the technique of gas chromatography coupled with mass spectrometry (Gas Chromatography–Mass Spectrometry, GC-MS). The analysis was performed on an Agilent Technologies Gas Chromatograph 7820A, connected to an Agilent Technologies 5977B mass spectrometer system based on electron impact (EI) and 70 eV of ionization energy. The gas chromatograph is equipped with a split/splitless injector and a capillary column HP5MS of 30 m, internal diameter of 0.25 mm, and membrane thickness of 0.25 μm. The temperature was programmed from 100 to 300 °C at a rate of 5 °C/min. The carrier gas was He at a flow rate of 0.7 mL/min, injection volume of 2 μL, split ratio of 1:10, and injector temperature of 280 °C. The identification was accomplished using computer searches on Wiley mass spectral databases (and database created from our research team). The components of propolis extract were determined by considering their areas as percentage of the total ion current.

### 4.5. Isolation of Compounds

Seven compounds were isolated by the technique of Preparative Thin-Layer Chromatography (20 cm × 20 cm glass plates, silica gel F_254_ Merck in CH_2_Cl_2_:MeOH 98:2) and their structure was identified by the technique of Nuclear Magnetic Resonance (NMR).

^1^H-NMR spectra were obtained on a Bruker DRX 400 instrument (400 MHz) using CDCl_3_ as a solvent and TMS as an internal standard.

From sample PP2 (40 mg), a mixture of pinocembrin and sakuranetin (2.2 mg) and pinostrobin (2.4 mg) was isolated:

Pinocembrin (1): (400 MHz, CDCl_3_) δ: 2.83 (1H, dd, *J* = 3, 17 Hz, Ha-3), 3.09 (1H, dd, *J* = 17, 3 Hz, Hb-3), 5.43 (1H, dd, *J* = 3, 13 Hz, H-2), 6.00 (2H, dd, *J* = 3, 0.7 Hz, H-6, H-8), 7.42 (5H, m, Ar) (Appendix A).

Sakuranetin (2): (400 MHz, CDCl_3_) δ: 2.79 (1H, dd, *J* = 17.6, 3.2 Hz, Ha-3), 3.10 (1H, dd, *J* = 17.6, 3.2 Hz, Hb-3), 3.83 (3H, s, 7-OMe), 5.38 (1H, dd, *J* = 13.2, 3.2 Hz, H-2), 6.10 (1H, d, *J* = 2.4 Hz, H-6), 6.07 (1H, d, *J*= 2.4 Hz, H-8), 6.91 (2H, d, *J* = 8.8 Hz, H-3′, H-5′), 7.36 (2H, d, *J* = 8.8 Hz, H-2′, H-6′) (Appendix A).

Pinostrobin (3): (400 MHz, CDCl_3_) δ: 2.83 (1H, dd, *J* = 17.1, 3.4 Hz, Ha-3), 3.10 (1H, dd, *J* = 17.1, 3.4 Hz, Hb-3), 3.82 (3H, s, 7-OMe), 5.43 (1H, dd, *J* = 13, 2.8 Hz, H-2), 6.08 (2H, dd, *J* = 2.3, 5.8 Hz, H-6, H-8), 7.45 (5H, m, Ar) (Appendix A).

From sample PP3 (40 mg), 7-oxo-dehydroabietinol (2.5 mg), dehydroabietic acid (3.0 mg), and a mixture of *Ε*-communic acid and *Z*-communic acid (4.1 mg) were isolated:

7-oxo-dehydroabietinol (4): (400 MHz, CDCl_3_) δ: 7.86 (1H, d, *J* = 2 Hz), 7.39 (1H, dd, *J* = 8, 2 Hz), 7.30 (1H, d, *J* = 8 Hz), 3.44 (1H, d, *J* = 10.8 Hz), 3.19 (1H, d, *J* = 10.8 Hz), 2.92 (1H, m, *J* = 6.9 Hz), 1.31 (3H, m), 1.26–1.24 (6H, 2 × d, *J* = 6.9 Hz), 0.92 (3H, brs) (Appendix A).

Dehydroabietic acid (5): (400 MHz, CDCl_3_) δ: 7.16 (1H, d, *J* = 8 Hz, H-11), 7.00 (1H, dd, *J* = 2, 8 Hz, H-12), 6.88 (1H, brs, H-14), 2.80–2.90 (3H, m, H-15, H-7), 2.21–2.25 (1H, dd, H-5), 1.28 (3H, s, Me-19), 1.23 (3H, s, Me-20), 1.24 (6H, d, *J* = 6 Hz, Me-16, Me-17) (Appendix A).

*Ε*-communic acid (6): (400 MHz, CDCl_3_) δ: 6.31 (1H, dd, *J* = 10.3, 17.3 Hz, H-14), 5.41 (1H, t, *J* = 6.4 Hz, H-12), 5.05 (1H, d, *J* = 17.3 Hz, Ha-15), 4.88 (1H, d, *J* = 10,7 Hz, Hb-15), 4.84 (1H, s, Ha-17), 4.47 (1H, s, Hb-17), 1.75 (3H, s, Me-16), 1.26 (3H, s, Me-18), 0.66 (3H, s, Me-20) (Appendix A).

*Z*-communic acid (7): (400 MHz, CDCl_3_) δ: 6.80 (1H, dd, *J* = 10.3, 17.3 Hz, H-14), 5.31 (1H, t, *J* = 6.5 Hz, H-12), 5.18 (1H, d, *J* = 17.3 Hz, Ha-15), 5.09 (1H, d, *J* = 10.9 Hz, Hb-15), 4.47 (1H, s, Ha-17), 4.84 (1H, brs, Hb-17), 1.77 (3H, s, Me-16), 1.28 (3H, s, Me-18), 0.66 (3H, s, Me-20) (Appendix A).

### 4.6. Antimicrobial Bioassay

All extracts were investigated for their antimicrobial activity, against 8 strains of human pathogenic bacteria and 3 fungi. In vitro antibacterial studies, first, were carried out by the disc diffusion method [5] measuring the zone of inhibitions (mm) against the two Gram-positive bacteria: *Staphylococcus aureus* (ATCC 25923) and *S. epidermidis* (ATCC 12228); the four Gram-negative bacteria: *Escherichia coli* (ATCC 25922), *Enterobacter cloacae* (ATCC 13047), *Klebsiella pneumoniae* (ATCC 13883), and *Pseudomonas aeruginosa* (ATCC 227853); as well as the pathogen fungi *Candida albicans* (ATCC 10231), *C. tropicalis* (ATCC 13801), and *C. glabrata* (ATCC 28838). Furthermore, the Gram-positive oral bacteria: *S. mutans* and *S. viridans* (clinical strains), were also tested. All studied samples dissolved in dimethyl sulfoxide (DMSO) were screened for in vitro antibacterial and antifungal activities on Mueller-Hinton or Sabouraud broths as previously described [5].

First, the assays were carried out by the disc diffusion method measuring the zone of inhibitions. For each experiment, control disks with pure solvent were used as a blind control. Petri dishes had been previously inoculated with the tested microorganisms to give a final cell concentration of 107 cells/mL. Ten-microliter volumes of the above solutions were required to wet (impregnate) the test paper discs. The incubation conditions used in experiments were 24 h at a temperature of 37 °C. The growth conditions and the sterility of the medium of each strain were controlled and then the plates were incubated. The results were reported as the diameter of the zone of inhibition around each disk (in mm).

The minimal inhibitory concentrations (MICs) of the tested extracts were evaluated by the broth micro-dilution method. The sterile 96-well polystyrene microtitrate plates were prepared by dispensing 100 µL of the appropriate dilution of the tested extracts in a broth medium, per well, in order to obtain the final concentrations of the tested extracts that ranged from 0.50 to 10 mg/mL. The inoculums that were prepared with fresh microbial cultures in sterile 0.85% NaCl, to match the turbidity of the 0.5 McFarland standard, were added to the wells to obtain a final density of 1.5 × 10^6^ CFU/mL for bacteria and 5 × 10^4^ CFU/mL for yeasts (CFU: colony forming units). After incubation (37 °C for 24 h), the MICs were assessed visually for the lowest concentration of the extracts, showing the complete growth inhibition of the reference microbial strains. An appropriate DMSO control (at a final concentration of 10%), a positive control (containing the inoculum without the tested samples), and the negative control (containing the tested derivatives without the inoculum) were included on each microplate.

Standard antibiotic netilmicin (at concentrations of 4–88 μg/mL) was used in order to control the sensitivity of the tested bacteria and sanguinarine for oral bacteria, whilst 5-flucytocine and itraconazole (at concentrations of 0.5–25 μg/mL), as well as amphotericin B, were used as controls against the tested fungi (Sanofi, Diagnostics Pasteur at concentrations of 30, 15, and 10 μg/mL). For each experiment, any pure solvent used was also applied as a blind control. The experiments in all cases were repeated three times and the results were expressed as mean values.

### 4.7. Total Phenolic Content

The total phenolic content of the samples was determined using the Folin-Ciocalteu method [39]. In a 96-well plate, 25 μL of propolis extracts (4, 2, and 1 mg/mL) or a standard solution of gallic acid (2.5, 5, 10, 12.5, 20, 25, 40, 50, 80, and 100 g/mL) in dimethylsulfoxide (DMSO) was added to 125 μL of a Folin-Ciocalteu solution (10%), followed by the addition of 100 μL of 7.5% sodium carbonate. The plate was incubated for 30 min, in darkness, at room temperature. The absorbance at 765 nm was measured using a TECAN Infinite m200 PRO multimode reader (Tecan Group, Männedorf, Switzerland). All measurements were performed in triplicate, the mean values plotted on a gallic acid calibration curve, and the total phenolic acid content expressed as mg Equivalent to Gallic Acid (GAE) per gram of dry extract.

### 4.8. DPPH Assay

For the determination of the antioxidant activity, propolis extracts (4, 2, and 1 mg/mL) were prepared using DMSO, as a solvent. In a 96-well plate, 10 μL of each sample was mixed with 190 μL of DPPH solution (12.4 mg/100 mL in ethanol) and then incubated, at room temperature, for 30 min in dark room conditions. The absorbance was measured at 517 nm. All measurements were performed in triplicate and gallic acid was used as the positive control [39]. The % inhibition of the DPPH radical for each dilution was calculated using the following formula: %Inhibition = {[1 − (A − AB)]/AT} × 100, where A is the absorbance of the sample, AT is the absorbance of the control, and AB is the absorbance of the sample without the DPPH radical.

## 5. Conclusions

In conclusion, the investigated samples showed a European profile containing flavonoids and aromatic acid esters, with the exception of the sample from the Vrondero region, which showed a Mediterranean-type profile with the presence of diterpenes and the absence of both aromatic acid esters and flavonoids. This particular result could be explained due to the intensive presence of Juniper trees in this area, and a further proof of this is the detection of the diterpene 7-oxo-dehydroabietinol—for the first time from a propolis sample—in the plant species of *Juniperus oxycedrus* L. subsp. *deltoids* collected from this area. Pollen spectra of PNP propolis revealed a variety of not only nectariferous but also nectarless sources available to bees and included taxa whose presence indicated their Prespa region origin (*Quercus* sp., *Pinus* sp., *Campanula* sp. Campanulaceae, Cistaceae, and Poaceae). Regarding the amount and diversity of pollen present in propolis samples, it should also be emphasized that the PNP area is floristically rich and interesting due to its climate and geographical location.

All extracts showed a promising profile with a wide spectrum of antibacterial activity, against both Gram-positive and -negative bacteria and, moreover, moderate antifungal activity. The antioxidant activity of the majority of the samples was significantly high according to their phenolic content. The present study demonstrated that propolis originated from such a unique habitat as PNP, might be a rich source of antibacterial and antioxidant compounds, and is also an important natural product with plenty of beneficial potentials concerning human health. It is also worth mentioning that PNP’s flora of both melliferous and honeydew resources plays a very important role as a sustainable development factor of stable apiculture practice, with significant chemical diversity demonstrated, even in samples from a limited geographical area as this of PNP.

## Figures and Tables

**Figure 1 molecules-27-07018-f001:**
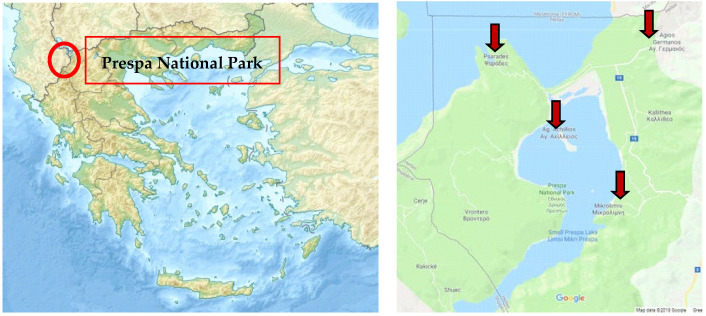
Map of PNP in Northwest Greece and collection areas.

**Table 1 molecules-27-07018-t001:** Chemical composition by GC-MS analysis of the propolis from the PNP (NW Greece).

RT (min)	Compound	PP1	PP2	PP3	PP4	PP5
16.03	succinic acid	1.77				0.54
20.13	decanoic acid/capric acid			0.46		
21.30	malic acid	0.82				0.22
22.04	unidentified sugar			1.16		
22.09	cinnamic acid	0.93			0.86	1.17
24.04	unidentified sugar	1.08	0.33			
24.14	p-hydroxybenzoic acid	0.26				0.24
24.33	unidentified sugar			1.07		0.17
24.60	dodecanoic acid/lauric acid			0.48		
24.00–26.00	unidentified sugars	2.01	2.98	0.21		
26.35	pentanedioic acid			0.37		0.38
27.65	p-coumaric acid		0.52			
27.90	nonanedioic acid/azelaic acid	1.13				
27.50–30.50	unidentified sugars	46.8	79.91	69.75	69.88	65.10
29.86	vanillylpropionic acid					0.38
30.1–30.6	unidentified sugars					6.97
30.60	4-hydroxycinnamic acid/p-coumaric acid	3.26	0.61		0.91	0.85
31.06–32.24	unidentified sugars	8.86	2.90		1.05	12.93
31.50	epi-manoyloxide			0.43		
32.09	palmitoleic acid			0.98		
32.20	3,4-dimethoxy cinnamic acid				1.29	
32.25	unidentified sugar			3.61		
32.49	3-hydroxymyristic acid	0.37				
32.51	gluconic acid			1.63		0.41
32.55	unidentified sugar	0.37		0.32		
32.62	palmitic acid	0.75	0.98	2.12	2.74	0.80
32.8	unidentified sugar					0.27
33.30	ferulic acid	2.86	0.68		0.53	0.68
33.57	isoferulic acid	2.70				
34.17	unidentified sugar	0.38	0.23			
34.48	caffeic acid	3.53	1.15		0.69	1.06
35.40	pentenyl p-coumarate	0.25				
35.47	unidentified sugar					0.29
35.50	linoleic acid			0.78	<0.4	
35.62	oleic acid	1.30	0.68	2.36	1.66	0.48
35.81	unidentified sugar	0.92	0.40			
35.90	diterpenic acid			<0.09		
36.24	ferruginol			0.41		
37.01	pimaric acid			1.41		
37.11	abietic acid			0.36		
37.18	pentenyl ferulate	0.30				
37.43	unidentified sugar	0.28				
37.35	isopimaric acid			2.08		
37.45	communic acid			1.85		
37.55	totarol			0.55		
37.93	pentenyl ester of isoferulic acid	0.31				
38.10	3-methyl-3-butenyl ester of caffeic acid	3.23	1.06		1.39	0.57
38.24	3-methyl-6-butenyl of ferulic acid					0.24
38.33	pinostrobin		0.34			
38.36	dehydroabietic acid			0.43		
38.75	2-methyl-2-butenyl ester of caffeic acid	0.91	0.33		<0.4	<0.2
38.97	3-methyl-2-butenyl ester of caffeic acid	2.95	1.02		0.76	0.63
39.12	isoagatholal			<0.09		
39.45	caffeic acid ester	0.27				
40.16	pinostrobin chalcone	0.28	0.99			
40.71	pinocembrin chalcone	0.95				0.25
40.82	pinocembrin	1.06				
40.96	agathadiol + imbricataloic acid			2.37		
41.33	pinobanksin chalcone	0.46				
41.58	isocupressic acid			0.42		
41.75	pinobanksin	0.80				
42.06	3-acetyl alpinon				1.09	
42.64	sakuranetin				0.85	
42.92	pinobanksin 3-o-acetate	0.50				
43.38	benzyl ester of caffeic acid	0.35	1.02		3.37	0.07
43.71	chrysin	0.52			1.96	0.60
44.03	galangin	0.68				
44.53	phenylethyl ester of caffeic acid	0.39	0.37		0.73	0.32
47.64	cinnamyl ester of caffeic acid	0.30	0.23			0.53
51.02	triterpene				1.58	

**Table 2 molecules-27-07018-t002:** Categories of chemical compounds of the propolis from the PNP (NW Greece).

	PP1	PP2	PP3	PP4	PP5
Aliphatic acids (%)	6.14	1.66	9.18	4.40	3.21
Phenolic acids (%)	13.54	2.96	-	4.28	4.00
Phenolic acid esters (%)	9.26	4.03	-	6.25	2.36
Diterpens (%)	-	-	10.31	-	-
Triterpens (%)	-	-	-	1.58	-
Flavonoids and chalcones (%)	5.25	1.33	-	3.90	0.85
Unidentified sugars (%)	60.70	86.75	76.12	70.93	85.73
Other unknown compounds (%)	4.16	2.96	3.27	8.67	1.60

**Table 3 molecules-27-07018-t003:** Pollen grain analysis of the propolis from the PNP (NW Greece).

	PP1	PP2	PP3	PP4	PP5
Secondary pollen (16–45%)	*Centaurea* sp. Asteraceae 27% Liliaceae 25% *Pyrus/Prunus* Rosaceae 24%	*Trifolium* sp. Fabaceae 25% *Pyrus/Prunus* Rosaceae 25% *Centaurea* sp. Asteraceae, 16%	Liliaceae 26% *Trifolium* sp. Fabaceae 24% Boraginaceae 22%	Liliaceae 28% *Trifolium* sp. Fabaceae 25%	Asteraceae 28% *Trifolium* sp. Fabaceae 27%
Important minor pollen (3–15%)	Asteraceae, 8% Apiaceae 5%	Asteraceae 14% Apiaceae 11%	*Centaurea* sp. Asteraceae 15% *Centaurea cyanus* Asteraceae 8%	*Pyrus/Prunus* Rosaceae 13% Asteraceae 12% Rosaceae 9% *Centaurea cyanus* Asteraceae 7%	*Pyrus/Prunus* Rosaceae 14% Rosaceae 14% Liliaceae 12%
Minor pollen (<3%)	*Brassica* sp. Brassicaceae 2% *Trifolium* sp. Fabaceae, 2% Boraginaceae, 2% *Vicia* sp. Fabaceae, 2% *Castanea sativa* Fagaceae 1%	*Castanea sativa* Fagaceae 2% Liliaceae 2% *Ononis* sp. Fabaceae 2% *Brassica* sp. Brassicaceae 1%	Asteraceae 2% Apiaceae 1% *Pyrus/Prunus* Rosaceae 1%	*Centaurea* sp. Asteraceae 2% Boraginaceae 2%	Apiaceae 3%
Pollen from nectarless plants/Isolated pollen	*Quercus* sp. Fagaceae, *Chenopodium* sp. Chenopodiaceae, *Scabiosa* sp. Dipsacaceae, *Pinus* sp. Pinaceae, Poaceae, *Euphorbia* sp. Euphorbiaceae, *Campanula* sp. Campanulaceae, *Oxalis* sp. Oxalidaceae Cistaceae Caryophyllaceae *Geranium* sp. Geraniaceae, *Polygonum aviculare* Polygonaceae *Populus* sp. Salicaceae	*Salix* sp. Salicaceae *Quercus* sp. Fagaceae, Cistaceae *Campanula* sp. Campanulaceae, *Vicia* sp. Fabaceae, *Pinus* sp. Pinaceae, *Chenopodium* sp. Chenopodiaceae, *Scabiosa* sp. Dipsacaceae, *Geranium* sp. Geraniaceae, Lamiaceae, Caryophyllaceae *Carex* sp. Cyperaceae Cupressaceae Poaceae, Cistaceae *Tribulus* sp. Zygophyllaceae *Ephedra* sp. Ephedraceae,	Cupressaceae *Ephedra* sp. Ephedraceae, *Smilax* sp. Smilacaceae *Salix* sp. Salicaceae, *Quercus* sp. Fagaceae, *Papaver rhoeas* Papaveraceae *Campanula* sp. Campanulaceae, *Vicia* sp. Fabaceae, Caryophyllaceae Cistaceae *Buxus semprervirens* Buxaceae Plantaginaceae, *Fraxinus* sp. Oleaceae, *Onobrychis* sp. Fabaceae *Populus* sp. Salicaceae, Rosaceae Poaceae, *Brassica* sp. Brassicaceae *Euphorbia* sp. Euphorbiaceae	*Quercus* sp. Fagaceae, Cistaceae *Campanula* sp. Campanulaceae, *Pinus* sp. Pinaceae, Poaceae, Cupressaceae Brassica sp. Brassicaceae *Chenopodium* sp. Chenopodiaceae, *Malva* sp. Malvaceae *Geranium* sp. Geraniaceae, Lamiaceae *Verbascum* sp. Scrophulariaceae *Loranthus europaeus* Loranthaceae, *Euphorbia* sp. Euphorbiaceae	*Quercus* sp. Fagaceae, *Pinus* sp. Pinaceae, *Populus* sp. Salicaceae Cupressaceae *Fraxinus* sp. Oleaceae, *Campanula* sp. Campanulaceae, *Euphorbia* sp. Euphorbiaceae Poaceae, *Papaver rhoeas* Papaveraceae, Scrophulariaceae *Acer* sp. Aceraceae *Artemisia* sp. Asteraceae, *Rhamnus* sp. Rhamnaceae

**Table 4 molecules-27-07018-t004:** Antimicrobial activities (zones of inhibition in mm/ and MIC mg/mL, *n* = 3).

Samples	*S. aureus*	*S. epidermidis*	*P. aeruginosa*	*K. pneumoniae*	*E. cloacae*	*E. coli*	*S. mutans*	*S. viridans*	*C. albicans*	*C. tropicalis*	*C. glabrata*
PP1	15/0.96	15/0.88	12/1.87	13/1.52	12/1.36	12/1.42	13/1.23	13/1.25	10/2.38	12/1.53	12/1.30
PP2	13/1.27	14/1.15	12/1.80	12/1.79	12/1.28	12/1.35	12/1.54	12/1.50	10/2.17	12/1.40	12/1.27
PP3	15/0.85	17/0.56	13/1.00	13/1.12	13/0.98	14/0.85	15/0.77	14/0.90	11/1.20	12/1.13	12/1.15
PP4	12/1.55	13/1.30	12/1.77	12/1.65	12/1.27	12/1.45	12/1.25	12/1.3	10/2.24	11/1.32	12/1.25
PP5	13/1.78	13/1.85	12/1.95	12/1.70	12/1.60	11/1.90	12/1.48	12/1.59	10/2.40	11/1.41	11/1.37

**Table 5 molecules-27-07018-t005:** TPC and antioxidant activity as DPPH % inhibition for propolis samples.

Samples	TPC mg/g Extract	DPPH (% Inhibition)		
		200 μg/mL	100 μg/mL	50 μg/mL
PP1	203.77 ± 1.04	90.87 ± 0.23	83.25 ± 1.14	44.72 ± 0.46
PP2	192.23 ± 3.39	91.31 ± 0.22	77.83 ± 1.79	42.32 ± 3.50
PP3	39.23 ± 0.36	16.69 ± 0.97	8.97 ± 0.44	3.23 ± 0.33
PP4	153.82 ± 1.45	86.74 ± 0.08	47.03 ± 0.83	27.12 ± 0.50
PP5	173.37 ± 1.79	90.94 ± 0.07	74.54 ± 1.83	39.17 ± 2.53

**Table 6 molecules-27-07018-t006:** Propolis collection from stable beehives of PNP.

Sample	Collection Area	Collection Season
PP1	Agios Germanos	Summer 2019
PP2	Agios Germanos	Summer 2020
PP3	Vrondero	Summer 2020
PP4	Latsista	Summer 2020
PP5	Agios Achillios	Summer 2020

## Data Availability

Not applicable.

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
