# Peer review of "Qualitative Analysis Related to Palynological Characterization and Biological Evaluation of Propolis from Prespa National Park (Greece)"

_molecules, 2022, doi:10.3390/molecules27207018_

Round 1

Reviewer 1 Report

In general, I think it is a good article. However, there are several aspects that are not fully clarified. In the attached file, the observations are detailed in messages and highlighted in yellow.

Author Response

  • Thank you for all your suggestions. All the required changes in the highlighted text, have been accepted and introduced.
  • Figure 1 was deleted from the manuscript and it was included in Supplementary file
  • The detailed method for pollen analysis has been added in Material and Methods Section 4.2 Pollen analysis
  • The detailed Method for antimicrobial assays has been added in Material and Methods Section 4.6 Antimicrobial bioassay
  • Zone of inhibition in mm (added in the table title and in the section 4.6 Antimicrobial bioassay) and MICs in mg/mL
  • The metabolites described have been isolated and structurally determined in order to be further used as internal standards in our continuous and systematic studies on propolis, from different countries all over the world, where these could be used for standardization as well as comparisons.
  • As previously referred the plan for these study was to isolate and use as internal standards while in a forthcoming draft manuscript a panel of several diterpenes (among which the refered once) will be screened and further evaluated for their bioactivities in comparison with literature data. It has to be also noticed that a couple of diterpenes refered in this study have been already screened previously by our team for their activities (Popova et al Phytochemistry 2009; https://reader.elsevier.com/reader/sd/pii/S0031942209003008?token=A0502B3150683738FE76B72F09D9A662AED9D52911FCD18A862860ECDEA22FC98D444261EE25646D336CBAA0BAC825CE&originRegion=eu-west-1&originCreation=20221013092808)
  • The quantity of each compound was added in the section 4.5. Isolation of compounds
  • The J value for these germinal protons is 17Hz (please see file:///C:/Users/E73%20SFF/Downloads/molecules-23-00852-s001.pdf/ and https://etd.ohiolink.edu/apexprod/rws_etd/send_file/send?accession=osu1322496872&disposition=inline Page 73)

Reviewer 2 Report

This impressive paper presented a systematic characterization of propolis including their chemical component, metabolites, pollen grains as well as important properties (such as antibacterial, antifungi, and anti-oxidation). Based on the study results indicated that the geography and plant source could impact the generation of different types of propels. Also, the author used the data to prove the potential health benefit of propolis. The author may consider some minor clarifications and content reorganization to help the readers understand the content better. 

  1. In Page 4, the author used “aromatic acid” and “aromatic acid esters” in Table to while used “phenolic acids” in line 95. Since phenolic acid is one type of the aromatic acids, could the author clarify whether “aromatic acid” in Table 2 was used interchangeably as “phenolic acid” or not? If so, it might be helpful if the author could use consistent terms. 
  2. Page 5, shall figure S1 in Supporting Information section? Also, could the author show the spectral data to indicate the existence of the seven metabolites? One suggestion maybe  the author could consider to combine the data in section 4.4 to section 2.2, with NMR spectra (figure)
  3. Section 2.2, could the author clarify whether the percentage used in Table 3 mass based or molar based? 
  4. In Table 3, Considering PP1 and PP2 are from the same location, why they have some unique secondary pollen?For instance PP1 contains Liliaceae while PP2 contains Trifolium.
  5. Page 10, could the author clarify why different years were taken into account for sample from Agios Germanos, but not for sample from other locations?
  6. Section 2.4, are there any repeats for DPPH test? 
  7. Line328, “DDPH” seems a typo, should be “DPPH”

Author Response

  1. Followed 2nd Reviewer’s suggestions all terms changed to be consistent. The term “phenolic acids” is used
  2. Fig S1 following the suggestion was moved in Supplementary file. According Journal’s editing rules was not possible to introduce NMR’s data in 2.2. All the spectral data of the isolated compounds have been added in the Supplementary file
  3. In Table 3 in order to present the frequency as a percentage (%), at least 500 pollen grains were counted from two slides created separately from the same sample. Some pollen grains cannot be identified to the species level. In these cases, pollen grains can be classified at genus or family level. Pollen grains, depending on their frequency, are classified into the following categories:"Predominant pollen" (more than 45% of the pollen grains counted), "Secondary pollen" (16-45%), "Important minor pollen" (3-15%), "Minor pollen " (less than 3%). Pollen whose frequency is less than 1% is characterized as "isolated". Pollen grains from nectarless plants are simply recorded as present. All these info has been added in the section 4.2 Pollen analysis
  4. In Table 3, even though PP1 and PP2 are from the same location, they are not identical nevertheless the bee-hives are stable, as the climatological conditions in the area (and in each geographic area, to be honest) are not identical (rainfall, humidity, temperature etc). As the pollen analyses show the exact profile of the flora, it is obvious that this floristic profile changes annually due to the climate changes. In both samples, PP1 and PP2 both Liliaceae and Trifolium are present reflecting the flora of the area. The percentages are varied according to the beekeeping handles and the period of collecting the samples, which related mainly with the blooming period of the plants as well as the population of each plant species close to the bee hives each year.  
  5. When the study started, only propolis sample from Agios Germanos area was available  from the collection year 2019 (kept frozen, in appropriate conditions for analysis)
  6. All measurements for DPPH test were performed in triplicates and gallic acid was used as positive control as it is referred in section 4.8 DPPH assay
  7. Line328, the typo mistake has been corrected

Reviewer 3 Report

1. As a suggestion, the introduction should be reorganized. Try first to describe the theoretical framework of the manuscript and leave the objective or intention of the research at the end.

2. Despite mentioning it in the Abstract and Introduction, it is not described how the palynological characterization was done in the Methods section. This point must be added because it is even part of the title of the manuscript.

The process of analyzing the pollen grains should be described, detailing how they were isolated from the propolis samples and their identification by microscopical analysis. If it was the case, mention pollen identification databases of the European floral species consulted. Even mention if there are specific databases of the endemic species of the PNP collection area of northwest Greece.

3. Some observations and suggestions have been noted directly in the pdf of the manuscript.

Author Response

  1. Following the Reviewer’s suggestion the Introduction has been re-organised
  1. The detailed method for pollen analysis has been added in Material and Methods Section 4.2 Pollen analysis
  1. All suggestions and comments noted in the pdf of the manuscript were accepted and corrected accordingly
  • COV-SARS-19 has been replaced by the term SARS-CoV-2
  • Italics has been used in the terminology throughout the manuscript
  • The titles of the tables have been improved containing the required info
  • caproic acid has been corrected to capric acid
  • the term “unidentified sugars” has been used instead of “sugars” as well as the term” other unknown compounds” instead of “unknown”
  • Figure 1 was deleted from the manuscript and it was included in Supplementary file
  • the collection season was the summer period of the year (added in table 6 and section 4.1 Samples)
  • the abbreviation TPC has been added in section 2.4
  • the term “Picture” has been replaced by” Figure”
  • info regarding the extraction have been added in the section 4.3 Extraction and sample derivatization
  • More details regarding the chromatographic conditions have been added in 4.4. GC-MS analysis, rerarding the isolation procedure in section 4.5. Isolation of compounds and regarding the antimicrobial tests in section 4.6. Antimicrobial bioassay